# Seasonal Comparison of Microbial Hygiene Indicators in Raw and Pasteurized Milk and Cottage Cheese Collected across Dairy Value Chain in Three Regions of Ethiopia

**DOI:** 10.3390/foods12244377

**Published:** 2023-12-05

**Authors:** Henok Nahusenay, Alganesh Tola, Tesfaye Sisay Tessema, Jessie Vipham, Ashagrie Zewdu Woldegiorgis

**Affiliations:** 1Center for Food Science and Nutrition, College of Natural Sciences, Addis Ababa University, New Graduate Building, Addis Ababa P.O. Box 1176, Ethiopia; hanimaby@gmail.com; 2Food Science and Nutrition Research Directorate, Ethiopian Institute of Agricultural Research, Addis Ababa P.O. Box 036, Ethiopia; 3Holeta Agricultural Centre, Ethiopian Institute of Agricultural Research, Holeta P.O. Box 036, Ethiopia; alguto1999@gmail.com; 4Institute of Biotechnology, New Graduate Building, Addis Ababa University, New Graduate Building, Addis Ababa P.O. Box 1176, Ethiopia; tesfaye.sisayt@aau.edu.et; 5Department of Animal Science and Industry, Kansas State University, 247 Weber Hall, Manhattan, KS 66506, USA; jessiev@ksu.edu

**Keywords:** milk, cottage cheese, indicators bacteria, raw milk, seasonality, pasteurized milk

## Abstract

A longitudinal design with a simple random sampling method was used to collect and compare microbial hygiene levels between the dry season (January to April) and wet season (June to August). A total of 456 milk and cottage cheese samples were collected from each site along the dairy value chain from three regions. Enumeration of total aerobic mesophilic bacteria (APC), total coliforms (TCC), and Escherichia coli (EC) was performed according to standard methods. Independent *t*-tests were employed to assess the significant variation at (*p* < 0.05) between the two seasons. The cumulative result of APC of 7.61 log cfu/mL and g and TCC of 3.50 log cfu/mL in the dry season were significantly higher than the wet season of 7.15 log cfu/mL and 2.49 log cfu/mL, respectively, whereas generic *E. coli* count (EC) was significantly higher in the wet season (0.70 log cfu/mL and g) than that in the dry season (0.40 log cfu/mL and g). The results of hygienic indicator microbial load significantly varied with season. Hence, hygienic milk production and handling practices that comprehend seasonal influence should be implemented to improve the safety of milk.

## 1. Introduction

Dairy products, especially fresh milk, are regarded as a complete diet because they contain all the necessary nutrients [1]. Milk has a number of active chemicals that are important for both nutrition and health protection and is also a source of macro and micronutrients [2], such as fat and protein, which make it valuable both commercially and nutritionally [3]. Among the environmental factors, the type and safety of feeds the milking cows consume and the season of the year have a considerable influence on the safety of milk [4]. Foodborne illnesses, including milk-borne diseases, are a substantial public health concern for individuals and nations in the developed world and developing nations. This is because milk and other dairy products are produced in unclean conditions and using subpar production techniques [5]. Human diseases caused by milk-borne bacteria range from gastrointestinal disorders characterized by diarrhea and vomiting to other, more widespread, and even fatal foodborne infections. This issue extends beyond just public health issues and includes economic issues as well [6].

This is because milk is highly nutritious and is particularly sensitive to microbial deterioration [7]. Milk contamination occurs due to inappropriate sanitary practice, which occurs in all milk contact surfaces, including milkers’ hands, milk containers, and bulk tank containers [8]. In the wet season, the ground surface will become muddy, which favors the proliferation and transmission of pathogens [9]. In addition, in the wet season, prolonged precipitation occurs, which is mostly associated with the dissemination of microbes from contaminated areas, including water reservoirs like groundwater [10]. Unmanaged and spring groundwater is characterized by the absence of standard safety measures for restricting any foreign material, including contaminant dirt, from entering [11]. From our survey data (unpublished), most Ethiopian dairy farmers use groundwater for washing the teats of cows and milk storage utensils. During the dry season, temperature increases, creating a shortage of water, especially in rural parts of developing countries [12,13], where the water readily available to the farmers may not be enough to clean the equipment and the farm environment.

In Ethiopia, generally, there are three distinct seasons. From February to May, there is a short rainy season called the Belg. The long rainy season, known as the Kiremt, which lasts from late June to mid-September, comes next. The Bega, which typically lasts from October to January, is distinguished by relatively dry conditions across the majority of the nation and wet conditions and a secondary peak in rainfall over the south [14].

The survival and reproduction of microorganisms in food products are significantly influenced by seasonal fluctuations [15,16] which also influences both the quality and accessibility of food [17]. As a result, there are significant economic, societal, public, and environmental effects. Numerous studies have demonstrated how seasonal variations in temperature and humidity have an adverse impact on food safety and local and global food commerce [18]. Summer’s higher temperatures and humidity favor the development of bacteria and fungi in food products [19]. Different groups of microorganisms require different temperature ranges for optimum growth and multiplication. According to reports, bacteria, particularly those found in food, prefer a temperature range of 32 to 43 °C for growth [20]. In a recent study, several foodborne illnesses are more common in the summer than in the winter [21]. In the recent study conducted in the dry season by Mengstu et al. (2023) revealed that 86% and 90% of raw and pasteurized milk samples, respectively, collected from major milk shade of the Ethiopia Oromia, SNNP, and Amhara regions were substandard for microbiology requirement based upon Ethiopian Standards [22].

The objective of the current study was to detect and quantify the seasonal variation in microbial indicators of milk hygiene for raw milk, pasteurized milk, and cottage cheese across the value chain in the Oromia, SNNP, and Amhara regional states of Ethiopia.

## 2. Materials and Methods

### 2.1. Study Area and Study Design 

The investigation of seasonal variation of total aerobic mesophilic bacteria count (APC), total coliform count (TCC), and generic *E. coli* was conducted using a longitudinal study design. Three representative locations with the highest milk production capacity in each region were selected from the three regions, namely Debre-Zeit, Hawassa, and Bahir Dar from the Oromia, SNNP, and Amhara areas, respectively. 

### 2.2. Sample Collection, Transportation, and Storage 

A longitudinal study was conducted from January to April for the dry season and from June to August for the wet season in the same year of 2021 to assess the microbial hygiene of raw milk, pasteurized milk, and cottage cheese samples. A simple random sampling technique was used to collect milk samples across the milk value chain (i.e., producers, collectors, processors, and retailers). In the purposeful sampling technique used, milk samples were collected only from those value chain actors that are in the milk supply chain to which milk farmers deliver their milk-to-milk collection centers (MCCs), and the MCCs then delivered the collected milk in bulk to milk-processing factories; lastly, processed milk will be sold to retails. A total of 456 dairy product samples (184 raw milk, 184 pasteurized milk samples, and 88 cottage cheese) were collected from each site along the dairy value chain—namely, milk and cottage cheese producers (*n* = 136), milk collectors (*n* = 92), milk processors (*n* = 92), and pasteurized and traditional cottage retailers (n = 136) of the dairy value chain. Half of the total sample size, 228 dairy product samples, was collected in the dry season, and an equal sample size was then collected in the wet seasons from the sites of the regions. The sample from each site was: Debre-Zeit (*n* = 120), Hawassa (*n* = 60), and Bahir Dar (*n* = 48) from the Oromia, SNNP, and Amhara areas, respectively, in each season. Dairy product samples were collected aseptically using sterile containers and transported and a portable fridge was maintained at 4 °C. The samples arrived at the Addis Ababa University Center for Food Science Nutrition (CFSN) microbiology laboratory for microbial analysis. The microbial hygiene data published previously in the dry season was used for the seasonal comparison [22] with the wet season data collected here. 

### 2.3. Assessment of Microbiological Quality and Safety of Milk and Cottage Cheese 

#### 2.3.1. Total Aerobic Mesophilic Bacteria Count (APC)

The aerobic plate count (APC) was applied to enumerate total aerobic mesophilic bacteria in milk and cottage cheese samples by following the protocol for the pour plate method recommended by the Bacteriological Analytical Manual [23] and as outlined in detail in our previous publication [22]. 

#### 2.3.2. Enumeration of Coliform and *E. coil*

Enumeration of total coliforms and generic *E. coli* from milk and cottage cheese samples were performed on *E. coli*/coliform count (ECC) Petri film following the 3M manufacturer’s protocol [24] and as outlined in detail in our previous publication [22].

### 2.4. Statistical Analysis 

All microbial testing of the three regions were performed in duplicate analysis and the data were expressed as mean ± SD (Appendix A). Independent *t*-tests were employed to compare the significant variation between the dry and wet seasons of the microbial hygienic indicators’ levels). SPSS version 22 was used, and α = 0.05 was considered to be significant. For quantification data, sample type served as the experimental unit (raw milk, pasteurized milk, and cottage cheese), and data were analyzed utilizing a general linear mixed model with three × three factorial (region × value chain) design. The response variable was log cfu/mL for (raw milk and pasteurized milk) or log cfu/g (for cottage cheese), and the linear predictors included the fixed effects of the region (SNNP, Oromia, Amhara), value chain (producer, collector, retailer), and all two-way interactions.

## 3. Results

### 3.1. Total Aerobic Mesophilic Bacteria Count (APC) in Milk and Cottage Cheese Collected from the Study Sites

The percentages of milk and cottage cheese samples that meet the requirement for total aerobic bacterial count according to Ethiopian Standard Authority for milk and Kenya Bureau of Standard for cottage cheese are put forth in Table 1. The wet season investigation for APC count revealed that 100% of the collected samples from the three regions failed to meet the standards [25,26], whereas in our prior work [22], during the dry season in the Oromia region, 14.58% (*n* = 7) of the collected samples complied with the standards as shown in Table 1.

The APC in log cfu/mL of the pasteurized milk was significantly lower than that of raw milk in the three study sites, as shown in Table 2. During the wet season, higher values of APC of raw milk, pasteurized milk, and cottage cheese were observed in the SNNP region—9.67 ± 0.52 log cfu/mL, 6.976.17 ± 0.63 log cfu/mL, and 7.34 ± 0.32 log cfu/mL, respectively—comparted to the remaining study regions, as mentioned in Table 2.

Illustrated in Figure 1 is a seasonal comparison of APC in milk (log cfu/mL) and cottage cheese (log cfu/g) samples collected from the three study sites. In contrast with the raw milk sample, the APC loads in pasteurized milk and cottage cheese were found to be statistically higher in the dry season. Furthermore, the commutative APC load of the three-sample type still showed that the APC load was significantly higher in the dry season at *p* < 0.05.

### 3.2. Total Coliforms in Milk and Cottage Cheese Collected from the Study Sites

TCC of raw milk revealed that the samples that were gathered from the SNNP region (54.17%) mostly fit the standards stated by the Ethiopian Standard Authority (ESA, 2021) compared to the Oromia (33.3%) and Amhara (5%) regions in the wet season investigation, whereas the samples that were collected in the Oromia region majorly fit the standard during dry season regions as shown in Table 3. The level of TCC in pasteurized milk that was collected in the Oromia region during wet season revealed that the majority of the samples (97.92%) comply with the ESA (2021) compared to the SNNP (20.83%) and Amhara regions (0%), as mentioned in Table 3. The majority of cottage cheese samples that were collected in the wet season were shown to be complied with the Kenyan Standard for Cottage Cheese [26], while in the dry season, 58.33% of cottage cheese samples failed to comply with the standard as it was reported in our prior work [22]. 

Mean comparisons of total coliform counts of raw milk, pasteurized milk and cottage cheese samples during wet and dry seasons collected from the study region indicated in Table 4. The total coliform count was significantly (*p* < 0.05) higher in raw milk compared to pasteurized milk in all study regions during the wet season. 

Like raw milk, the pasteurized milk samples that were collected during the dry season (4.50 ± 2.1 log cfu/mL) showed strongly significantly higher TCC compared to the wet season (0.04 ± 0.1 log cfu/mL) in the Oromia region. Pasteurized milk that was collected during the wet season (3.71 ± 0.86 log cfu/mL) had a significantly higher TCC level than the dry season in the Amhara region. The TCC level variability in pasteurized milk due to season still had no significant impact on the samples that were collected from the SNNP region. The variation in TCC in the wet and dry seasons in cottage cheese samples had a similar pattern to that of raw and pasteurized milk in the three regions, as shown in Table 4. The variation in TCC levels in raw milk samples that were collected during the wet and dry seasons did not vary significantly in the three regions (Figure 1), whereas pasteurized milk and cottage cheese samples collected during the dry season showed a statistically significant difference at (*p* < 0.05) as compared to the wet season.

### 3.3. Seasonal Comparison of Generic Escherichia coli Count in Milk and Cottage Cheese Samples Collected from Three Regions 

Seasonal comparison of *Escherichia coli* count in milk and cottage cheese samples collected from three regions is presented in Table 5. The *E. coli* counts for wet season investigation revealed that 56.3% (n = 27) raw milk, 60.42% (n = 29) pasteurized, and 100% (n = 24) cottage cheese from the Oromia region; 41.6% (n = 10) raw milk, 66.67% (n = 16) pasteurized milk, and 100% (n = 12) cottage cheese from the SNNP region; and 85% (n = 17) raw milk, 35% (n = 7) pasteurized milk, and 100% (n = 8) from the Amhara region comply with the safety recommendation of the ESA (2021), which is to be nil in marketable dairy products [25].

Table 5 demonstrates the seasonal comparison of Escherichia coli (log cfu/mL org) count in milk and cottage cheese samples collected from the three regions. The *E. coli* counts of raw and pasteurized milk samples collected from SNNP and Amhara regions during the wet season were significantly different at (*p* < 0.05). In the SNNP region, raw milk had a higher *E. coli* count (1.48 ± 1.48 log cfu/mL), whereas in the Amhara region, pasteurized milk (1.77 ± 1.44 log cfu/mL) had a significantly higher load of *E. coli* bacteria. The counts of raw and pasteurized milk samples collected from the Oromia region did not vary significantly from each other.

Pasteurized milk samples that were collected from the three study sites had significant variation in *E. coli* counts, as illustrated in Figure 1, due to seasonal fluctuation. Unlike the cumulative TBC and TCC seasonal variations, these findings’ *E. coli* loads were found to be higher during the wet season.

### 3.4. Total Aerobic Mesophilic Bacteria Count (APC), Total Coliform Count (TCC), and Generic E. coli in Milk and Cottage Cheese across the Dairy Value Chain of Study Regions 

Total aerobic mesophilic bacteria count load was significantly higher in the wet season in both the producer (8.62 ± 0.35 log cfu/mL) and collector (8.55 ± 0.38 log cfu/mL) value chains of raw milk compared to the dry season, which was 7.53 ± 1.24 log cfu/mL and 7.31 ± 0.91 log cfu/mL, respectively, in the Oromia region. Contrary to the raw milk value chain actors, the APC was significantly higher during the dry season in both processers (5.85 ± 0.52 log cfu/mL) and retailers (6.44 ± 0.52 log cfu/mL) of the pasteurized milk value chain compared to the wet season sample collection from processors (5.46 ± 0.55 log cfu/mL) and retailers (5.31 ± 0.44 log cfu/mL). 

Total coliform count load variation due to season among the samples that were collected in the dairy value chain of Oromia region showed the TCC significantly higher in dry season in both milk types (raw and pasteurized) value chain actors. The TCC loads on those samples that were collected from the raw milk value chain were found to be 4.35 ± 2.09 log cfu/mL and 4.06 ± 2.7 log cfu/mL from the producer and collector value chains, respectively, during the wet season, whereas the TCC loads were found to be 6.27 ± 1.89 log cfu/mL (producer) and 5.77 ± 2.2 log cfu/mL (collector) during the dry season. The pasteurized milk that was collected during the wet season had 0.11 ± 0.44 log cfu/mL TCC from the processor value chain, and none of the samples that were collected from retailer value were found to have TCC, whereas in the dry season, 3.38 ± 1.89 log cfu/mL (processor) and 2.24 ± 2.5 log cfu/mL (retailer) total coliform counts were found. Unlike total coliform count in the dairy value chain, the *E. coli* load of the samples was significantly higher during the wet season. The raw milk that was collected from producers (0.78 ± 1.04 log cfu/mL) and collectors (1.23 ± 1.39 log cfu/mL) had *E. coli* in the wet season study. In the dry season, 0.13 ± 0.61 log cfu/mL and 0.27 ± 0.93 log cfu/mL *E. coli* loads were collected from the samples from the producer and collector value chains, respectively. *E. coli* counts in processor (0.91 ± 1.15 log cfu/mL) and retailer (0.90 ± 1.34 log cfu/mL) samples were significantly higher during the wet season than dry the season—0.08 ± 0.41 log cfu/mL and 0.15 ± 0.52 log cfu/mL, respectively.

The dairy value chain of this study, which was located in the SNNP region, revealed that APC was found to be significantly higher during the dry season in both sample types (raw and pasteurized) of value chain actors. The APC level in the raw milk value chain was found to be 11.05 ± 0.47 log cfu/mL and 10.56 ± 0.76 log cfu/mL in the producer and collector value chains, respectively, during the dry season, whereas during the wet season, values of 9.56 ± 0.51 log cfu/mL and 9.79 ± 0.53 log cfu/mL were found for the producer and collector, respectively. Furthermore, the APC load processer and retailer values were found to be 7.0 ± 1.15 log cfu/mL and 6.95 ± 0.37 log cfu/mL, respectively, while, in the dry season, 9.35 ± 1.72 log cfu/mL and 9.39 ± 1.7 log cfu/mL APC were found in the processor and retailer value chains, respectively. Seasonal attribution in TCC levels was found in the samples that were collected from the collector value chain of raw milk and the processor value chain of pasteurized milk, which revealed the TCC load to be significantly higher during the wet season. The total coliform count was found to be 5.31 ± 0.92 log cfu/mL in the collector and 7.01 ± 1.15 log cfu/mL in the processer value chain during wet season. In the dry season, TCC was found to be 3.00 ± 2.84 log cfu/mL and 4.5 ± 0.85 in the collector and processor value chains, respectively. Producers (3.69 ± 0.67 log cfu/mL) and retailer (2.33 ± 2.39 log cfu/mL) value chains did not have significant variation in total coliform count during the wet season compared to the dry season—the values were 4.10 ± 1.89 log cfu/mL and 1.11 ± 1.7 log cfu/mL, respectively. The raw milk samples that were gathered from the producer (0.75 ± 1.13 log cfu/mL) value chain and the pasteurized milk from the processer (1.17 ± 0.96 log cfu/mL) only showed a significant difference in *E. coli* count due to seasonal fluctuation compared to the other remaining value chain actors. The *E. coli* load was found to be higher in the dry season for the producer value chain, while for processer value chains, it was higher in the wet season.

The producer (9.11 ± 0.67 log cfu/mL) value chain of raw milk and the retailer (9.39 ± 1.7 log cfu/mL) value chain of pasteurized milk in the Amhara region were found to have statistically significant higher APC during the wet season than dry season 8.2 ± 0.67 log cfu/mL and 5.9 ± 0.18 log cfu/mL, respectively. The total bacterial counts from the sample collected from the collector (8.88 ± 0.21 log cfu/mL) and processor (5.71 ± 0.49 log cfu/mL) value chain did not have significant variations in APC when compared to those in the dry season—8.72 ± 0.83 log cfu/mL and 5.85 ± 0.16 log cfu/mL, respectively. All of the dairy value chains in the Amhara region for both milk sample types (raw and pasteurized) were found to have significantly higher TCC during wet season sample collection compared to dry season sample collection. High levels of the TCC were observed from the samples that were collected from a collector (5.91 ± 0.99 log cfu/mL) and a producer (5.41 ± 75 log cfu/mL) compared to the processor (3.97 ± 1.04 log cfu/mL) and retailer 3.45 ± 0.58 log cfu/mL) during the wet season. The same is true for the dry season higher—TCCs were found in those samples that were collected from producers (2.61 ± 3.41 log cfu/mL) and collectors (1.98 ± 1.07 log cfu/mL) than that those from the retailers (0.40 ± 1.27 log cfu/mL) and processors, which were nil. The *E. coli* load in the dairy value chains of Amhara region had no significant variation during wet and dry season sample collection. Total aerobic mesophilic bacteria count in the dairy value chain actors of the raw and pasteurized milk, located in the three study regions, revealed that only the samples that were collected from processer and retailer value chains had significant variation in both value chains; the APC was higher during dry season. As per the remaining value chains of the three regions, no significant difference was observed, as shown in Figure 2. Additionally, the cumulative APC in all value chains of raw and pasteurized milk showed no significant variation during the wet or dry seasons. 

As demonstrated in Figure 2, total coliform count varied significantly during the dry and wet seasons in samples collected from all value chain actors of raw and pasteurized milk of the study regions. The samples that were collected in the collector value chain only shown to have significantly higher TCC in the wet season, while the remaining value chain and the cumulative TCC of the all-value chain were shown to have a statistically higher TCC in the dry season as it put forth in Figure 2.

Seasonal comparison of *E. coli* count during wet and dry seasons in both milk type (raw and pasteurized) value chains revealed that the samples that were gathered from the collector value chain of the raw milk and the processer value chain of pasteurized milk were only found to have a significant difference in the *E. coli* load during the wet and dry seasons from the study regions, as shown in Figure 2. Furthermore, the cumulative *E. coli* load of all the dairy value chains of raw and pasteurized milk value chains was significantly higher during the wet season, as indicated in Figure 2.

The cottage cheese value chain is shown in Table 6. The seasonal variation of APC, collected during the dry season from both producers (6.05 ± 0.19 log cfu/g) and retailers (6.21 + 0.55 log cfu/g) in the Oromia region, retailers (9.0 ± 51 log10 cfu/g, 6.88 ± 0.2 log cfu/g) in the SNNP and Amhara regions, respectively, had significantly higher APC than the wet season. The cottage cheese that was collected from producers and retailers in the Oromia (2.24 ± 2.59 log cfu/g, 4.33 ± 3.48 log cfu/g) and Amhara (1.08 ± 1.25 log cfu/g, 2.09 ± 151 log cfu/g) regions was shown to have significantly higher TCC in the dry season compared to the wet season. Contrary to APC and TCC findings, no significant difference has been observed in *E. coli* levels during dry and wet season sample collections, as put forth in Table 6. 

The cumulative APCs in cottage cheese collected from the value chains from the three regions were found to have significant variation. In both value chains of the cottage cheese, APC statistically higher in dry season. Furthermore, APC was shown to be higher during dry season in the cumulative count of the producer and retailer value chains of cottage cheese. The total coliform count is significantly higher during the dry season in both value chains of cottage cheese and also in the cumulative count of TCC of both value chains. Unlike the APC and TCC results, no significant variation has been observed in the *E. coli* load of the cottage since none of the cottage cheese that was gathered in either value chains was found to have *E. coli*.

## 4. Discussion

### 4.1. Total Bacterial Count (APC) in Raw Milk, Pasteurized Milk, and Cottage Cheese 

The total bacterial count (APC) of raw milk collected from the three regions during the wet season (8.9 log cfu/mL) is higher than some previously reported studies in Ethiopia (Table 7). The reports from Debre Zeit by Solomon et al. (2013) showed a 7.07 log cfu/mL overall mean. In addition, Yirsaw et al. (2004) revealed that APC was found to be 8.13 log cfu/mL in the milk that was taken from the storage tank [27,28]. The studies by Weleregay et al. (2012) and Habtamu et al. (2018) from Hawassa also reported 7.28 log cfu/mL and 6.83 log cfu/mL TBC load in raw milk, respectively [29,30]. The TBC in raw milk that was gathered from Bahir Dar showed 7.61 and 8.12 log cfu/mL in individual farms and cooperatives, respectively, according to Tassew and Sefu (2011) [31]. Yeserah et al. (2019) and Takliye and Gizaw (2017) also reported 7.1 log cfu/mL and 6.87 log cfu/mL APC, respectively, in raw samples that were collected in Bahir Dar [32,33]. A similar report from the Oromia region by Gemechu (2016) also revealed 8.2 log cfu/mL APC in raw milk samples [34]. The higher APC value in the current study might be attributed to the variation in the sample size; variations in milking and hygienic practices followed in each region, and the present study covered three regions of the country and collected a large number of samples.

APC of pasteurized milk samples collected from three-study regions of 5.95 log cfu/mL for wet season was found to be lower compared to previously published works [35,36] and higher than [37] in different parts of Ethiopia (Table 7). Moreover, the APC load in pasteurized milk in the wet and dry seasons did not comply with the standards set by East African requirements (EAS 69:2006) and the Food and Drug Administration (FDA). The standards have declared the acceptable limit of TBC in pasteurized milk to be less than 3 × 10^4^ cfu/mL (4.47 log cfu/mL) and 2 × 10^4^ cfu/mL (4.3 log cfu/mL), respectively [23,38]. The higher APC load in pasteurized milk might be attributed to the absence of a cold chain, as the milk that was collected by the cooperative unions was transported without cooling over long distances to be delivered to processors. In addition, value chain effects could have contributed since the processer received the milk from both the collector and directly from the farmers. As this study showed that the APC load in raw milk is much higher, the pasteurization method may not be effective since the raw milk has too much APC. The other possible reasons could be that there might be faulty pasteurization or too poor pasteurization efficiency to kill the bacteria. From our risk factor survey (unpublished), it was observed that most of the milk-processing companies do not calibrate processing equipment and do not have certifications in good manufacturing practices.

Cottage samples that were collected during the wet season in the three study sites were found to have a 5.87 log cfu/g APC (Table 7). Comparative reports have been reported by Ashenafi (2006) (5.38 log CFU/g) and Solomon and Ketema (2011) (6.13 log CFU/g) [39,40]. Zelalem et al. (2005) and Mamo et al. (2016) also reported 8.8 log cfu/g and 6.5 log cfu/g APC in cottage cheese, respectively, from different areas in Ethiopia [41,42]. The overall mean of APC in cottage cheese in both seasons was above the tolerable limit for cooked cheese (4.7 log CFU/g) within the markets, as was mentioned by Al-Khatib and Al-Mitwalli (2009) [43].

**Table 7 foods-12-04377-t007:** Comparison of milk microbial load of the present study with some available data in the literature.

Milk Type	Microbial Indicator (log cfu/mL or g)			
	APC	Reference	TCC	Reference
Raw Milk	7.07	[27]	-	-
	8.13	[28]	-	-
	7.28	[29]	6.94	[30]
	6.83	[30]	8.58	[34]
	7.61	[31]	4.09	[35]
	7.1	[32]	3.59	[37]
	6.87	[33]	4.03	[44]
	8.2	[34]	5.47	[45]
	8.9	Present Study	4.59	Present Study
Pasteurized Milk	-	-	5.38	[30]
	6.14	[35]	6.20	[35]
	6.33	[36]	5.49	[36]
	2.42	[37]	6.05	[31]
	5.95	Present Study	1.53	Present Study
Cottage Cheese	5.38	[39]	-	-
	6.13	[40]	5.58	[39]
	8.8	[41]	4.4	[41]
	6.5	[42]	3.6	[42]
	4.7	[43]	4.42	[46]
	5.87	Present Study	0.09	Present Study

Seasonal variation of APC in raw milk samples was found to be significantly higher during the wet season, while some of the works from abroad report APC in raw milk to be higher during the wet season. Botton et al. (2019) and Simioni et al. (2014) from Brazil reported that the APC level was shown to be higher during the winter season compared to other seasons [47,48]. In another study by Scano and Caboni (2022) (Italy), APC in raw milk was higher during winter than in the spring and summer seasons [49]. Margatho et al. (2018) from Portugal found that APC count was higher in the wet season—winter (2.65) and autumn (2.62)—compared to summer (2.56) and spring (2.50) [50]. Even though microbe multiplication in most cases increases in warm and humid environmental conditions, in the rainy and cold seasons, cattle housed in intensive and semi-confinement systems arrive at the milking parlor with larger amounts of dirt on the udder [51]. This eventually increases the risk of cross-contamination of milk while milking [52]. A higher APC level during the dry season in raw milk has been reported by Elmoslemany et al. (2010) (Canada), who found that the APC levels in summer were 0.33 and 0.21 times higher than in fall and winter, respectively [53]. Hajmohamnadi et al. (2021) found the total bacteria count during summer (6.25 log cfu/mL) significantly higher than in winter (6.10 log cfu/mL) in raw milk collected from different parts of Iran [54]. In another similar study from Italy by Bertocchi et al. (2014), they revealed that the APC load in raw milk collected during summer (9.99 log cfu/mL) and spring (9.87 log cfu/mL) was higher than in fall (9.84 log cfu/mL) and winter (9.83 log cfu/mL) [55]. In warm and humid environments, the growth and number of environmental bacteria in cows’ bedding material increase owing to favorable temperature and humidity [54]. The bacteria most probably enter into dairy products in dairy farms that utilize traditional practices in countries like Ethiopia [56].

### 4.2. Total Coliform Count (TCC) in Raw Milk, Pasteurized Milk, and Cottage Cheese 

The TCC in raw milk samples collected from the three regions during the wet season (4.59 log cfu/mL) was found to be lower compared that in work in Ethiopia (Table 7) by Habtamu et al. (2018), Gemechu (2016), and Amakelwu et al. (2015) who reported 6.94 log cfu/mL, 8.58 log cfu/mL, and 5.47 log cfu/mL of TCC, respectively, but higher than Mirku et al. (2020) Tamirat (2018), and Mesfine et al. (2015), who found 4.09 log cfu/mL, 3.59 log cfu/mL, 4.03 log cfu/mL, and 1.82 log cfu/mL TCC, respectively [30,34,35,37,44,45]. 

The wet season TCC load in pasteurized milk (1.53 log cfu/mL) was found to be much lower than most of the previous reports in Ethiopia. Mikru et al. (2021), Habtamu et al. (2018), Tassew and Seifu (2011), and Aberra (2010) reported 6.20 log cfu/mL, 5.38 log cfu/mL, 6.05 log cfu/mL, and 5.49 log cfu/mL, respectively [30,31,35,36]. The reason for this decrement in TCC may be attributed to environmental factors since the above-mentioned study was done during the dry season. It may have contributed to the increment of total coliform load in pasteurized milk since in developing countries, cold chain transport and backup generators in pasteurized milk retailers are not common. Thus, such risk factors may have facilitated the multiplication of coliforms [57].

The TCC in cottage cheese samples that were collected from the three regions 0.09 log cfu/g (Table 7) was found to be lower than the earlier reports by Mamo et al. (2016), Yilma (2012), Ashenafi (2006), and Zelalem et al. (2005)—3.6 log cfu/g, 4.42 log cfu/g, 4.4 log cfu/g, and 5.58 log cfu/g, respectively [39,41,42,46]. The overall mean of TCC in cottage cheese of this study is below the standard stated by Kiiyukia (2003), which states that the TCC of heat-treated food should not be above (4 log10 cfu/g) [58].

Seasonality variation of total coliform count is not observed in raw milk; it is only observed in pasteurized milk and cottage cheese samples that were collected from the three regions. Seasonality of TCC in raw milk by Elmoslemany et al. (2016) from Saudi Arabia revealed that TCC levels were higher from November to January compared to other months of the year [53]. A similar study from Egypt by Zeinhom et al. (2016) indicated that TCC load is significantly higher in the months of summer (June–September) than in autumn (October–November) and winter (December–April) [59]. Zucali et al. (2011) also demonstrated TCC significantly higher in hot seasons (2.41 log cfu/mL) than in the months that have mild (1.92 log cfu/mL) and cold (1.6 log cfu/mL) temperatures [60].

The coliform counts in raw milk samples that were collected during summer (47.8%) had higher levels of coliforms those collected during winter (43.7), as reported by Salmn and Hamad (2011) from Sudan [61]. Contrary reports to the mentioned works have been released by Chanda et al. (2008). They found that the coliform load was higher in the wet months (June–August) 4.84 × 10^5^ than summer (4.11 × 10^5^), autumn (3.85 × 10^5^), and winter (2.75 × 10^5^) [62]. High rates of coliform load in the dry season compared to the wet season were due to the bacteria’s growth requirements. Coliforms are generally recognized due to their ability to ferment lactose to form gas and acid within 48 h at 32–37 °C and, more specifically, thermophilic ones can grow and ferment lactose at 44–45 °C [63], so they prefer the warm and humid environmental conditions of the dry season. The higher TCC load during the dry season in pasteurized milk might have arisen due to the absence of calibration of processing equipment and absence of food safety plans in most milk-processing factories. There is risk of biofilm production in processing equipment, especially in the process lines and joint parts [64], since most pasteurized-milk-processing factories in Ethiopia does not calibrate the equipment according to our unpublished survey data. In addition, the absence of cold chain transport might have also aggravated the problem since in the dry season, humid and warm temperatures prevail.

### 4.3. E. coli Count in Raw Milk, Pasteurized Milk, and Cottage Cheese 

One of the most common methods of detecting facial contamination is the *E. coli* count in dairy products [63]. The detection of *E. coli* during the wet season from the sample collection of three study regions was found to be higher in raw milk than that in pasteurized milk. A similar finding was reported by Habtamu et al. (2018); they discovered 26% of raw and 18.5% of pasteurized milk was tainted by *E. coli* [30]. In another report, Weleregay et al. (2012) found that 25% of pasteurized milk was contaminated by *E. coli,* which is lower than in the Oromia (35.98%) and Amhara (35%) regions [29]. Demme and Abegaz (2015) also detected *E. coli* in 18.6% of raw milk [65]. Higher detection of *E. coli* has been reported by Mikrus et al. (2021), Megersa et al. (2019), and Aberra (2010), who reported 60% in raw milk, 42% in raw milk, and 60% in pasteurized milk, respectively [35,36,66]. The pattern of *E. coli* detection in the Amhara region showed that the bacterium was predominantly detected in pasteurized milk compared to raw milk. This finding suggests that insufficient pasteurization, post-pasteurization problems, lack of cold chain during transportation, and lack of backup generators for pasteurized milk retail worsened the problem of safety of dairy products. The detection of *E. coli* in pasteurized milk was also reported by Iran by Vahedi et al. (2013)—of 100 pasteurized milk samples, 9 (9%) were positive for *E. coli* [67].

Seasonal variability of *E. coli* counts in dairy products has been reported by Zeinhom et al. (2016), who detected more *E. coli* in summer (June–September) than autumn (October–November) and winter (December–April) [59]. According to Zucali et al. (2011), *E. coli* contamination was higher in mild- and hot-temperature seasons than cold-temperature seasons [60]. Nevertheless, the present study found *E. coli* load to be significantly higher during the wet season than the dry season in the wet season, cows were significantly dirtier and more soiled as compared to the dry season; this was probably due to the difficulty in keeping cow beddings and passageways dry and clean during wet seasons and the consequent increased amounts of manure on the hindquarters, flanks, and udders [60]. Unhygienic conditions could be the main risk factor for the introduction of *E. coli* into dairy products which are produced at the farm level—most importantly, raw milk—since *E. coli* is an indicator of fecal contamination as it is mostly found in the intestines of warm-blooded animals [63]. Prolonged precipitation during the wet season was mostly associated with the dissemination of microbes from contaminated areas to water reservoirs like groundwater, which have unmanaged wells and are characterized mostly by the absence of standard safety measures for restricting any foreign material, including dirt entering into them [10,11]. From our parallel risk factor survey (unpublished) and observations during milk sampling, it was observed that most milk producers and collectors, and some milk processors use groundwater for cleaning, milking, storing, and processing equipment. Hence, in wet seasons, this groundwater and farm surfaces will receive microbes due to dissemination and anthropogenic activities. This eventually will worsen the problem by increasing the microbial load, including pathogenic load, on dairy products [68,69,70].

### 4.4. Cumulative APC, TCC and E. coli in Raw Milk, Pasteurized Milk, and Cottage Cheese

The cumulative APC of raw milk in producer (8.97 log cfu/mL) and collector (5.95 log cfu/mL) value chains of the wet season study showed a resemblance to the findings of Adugna et al. (2015), who found APC values of 7.6 ± 0.12 log cfu/mL and 7.56 ± 0.13 log cfu/mL from dairy farms (producer) and cooperatives (collector), respectively, from Bahir Dar, Zuria, and Mecha [71]. Contrary reports have been released by Berhe et al. (2020) from Tigray, which revealed that the APC mean value of raw milk collected from non-producer (non-farmer) value chains like cafeterias and hospitals (7.42 ± 0.272 log cfu/mL) was higher than from producers (farmers) 7.35 ± 0.18 log cfu/mL [72]. Welearegay et al. (2012) demonstrated that the APC of raw milk collected from distribution containers (collectors) (10.28 log cfu/mL) was higher than that collected from farms (5.93 log cfu/mL) [29]. Similar work by Amakelew et al. (2015) showed that raw milk that was collected from producers had lower APC (6.88 ± 0.3 log cfu/mL) than that of the collector (7.10 ± 0.79 log cfu/mL) [45]. This finding points out that the higher APC load in the dairy value chain has no seasonal effect and is most likely a systemic problem.

The present study also demonstrates that the cumulative TCC value of raw milk in the wet season that was gathered from the collectors (4.81 log cfu/mL) was higher than that from the producers (4.41 log cfu/mL). This finding was supported by Amakelew (2015), who reported a TCC in the collector value chain of 5.63 ± 0.56 log cfu/mL mean value compared to the producer (5.57 ± 0.22 log cfu/mL) value chain [45]. Similarly, Berhanu et al. (2020) reported that TCC load was lower in the distribution center (collector) (4.3 ± 0.5 log cfu/mL) than the farm (4.4 ± 0.5 log cfu/mL) [73]. In addition, Habtamu et al. (2018) reported the TCC of raw milk collected from dairy farms (6.63 ± 0.19 log cfu/mL) was found to be higher than that of raw milk collected from urban areas (7.07 ± 0.23 log cfu/mL). Habtamu et al. (2018) also reported that the TCC of pasteurized milk collected from the retailer shop was 5.87 ± 0.19 log cfu/mL, which is much higher than the cumulative TCC load in pasteurized milk that was collected from retailers from the three regions during the wet season (1.36 log cfu/mL) [30]. The higher TCC load in the collector value chain than producer may be attributed to the value chain effect since the collector value chain received raw milk from different farmers with different hygienic practices. This will create more chains to recover the microbes in those samples that were collected in the collection center. In the current study, the TCC was significantly higher during dry season, except for the collection centers. This may be because the coliforms belong to the enterobacteria family, which are recognized as mesophilic bacteria that prefer warm and humid conditions for growth and multiplication [63]. In addition, during the dry season, there is a shortage of water, more specifically in rural areas of Ethiopia, used to clean milk equipment and the farm environment [12,13].

The current result also demonstrated that the cumulative *E. coli* contamination in the dairy value chain of raw and pasteurized milk was significantly higher during wet season. The cottage cheese value chain predominantly reserves total aerobic mesophilic bacteria and total coliform during dry season This is due to unhygienic practices during open field production and retail and—most importantly—the open market, which is exposed to dust and insects. In addition, the environmental factors of humidity and warm temperature are more conducive to the growth and multiplication of microbes.

## 5. Conclusions

Among the milk sample types, raw milk samples were shown to have a significantly higher total bacterial load during the wet season. This justified proper heat treatment before the consumption of raw milk. The pasteurized milk and cottage significantly contained total aerobic mesophilic bacteria, total coliform, and *E. coli* during the dry season. It is mostly faulty pasteurization, improper handling, and the absence of a cold chain in addition to the humid warm temperature of the dry season that is conducive to the growth of most bacteria. This stresses the importance of maintaining standard safety procedures along the dairy value chain, most importantly during dry seasons. Proper storage temperature conditions must be followed while storing the milk, most importantly in the dry season, when heat and humidity are more common, creating conducive growth conditions for coliforms. Given that the dairy production practices at small dairy farms in Ethiopia are similar to those at small dairy farms in other African countries, the data reported in this paper will be relevant to Ethiopia and other countries with similar dairy farming situations and may have a positive impact.

## Figures and Tables

**Figure 1 foods-12-04377-f001:**
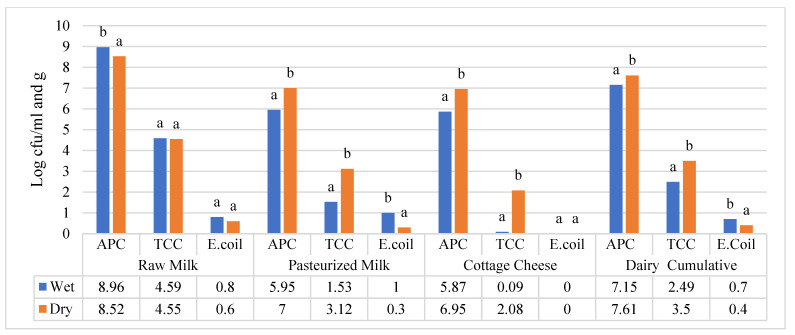
Comparison of total bacterial count (APC), total coliform count (TCC), and *E. coli* (EC) between dry and wet season in raw milk, pasteurized milk, and cottage cheese. Different superscripts letters above of the bar graph are significantly different at (*p* < 0.05).

**Figure 2 foods-12-04377-f002:**
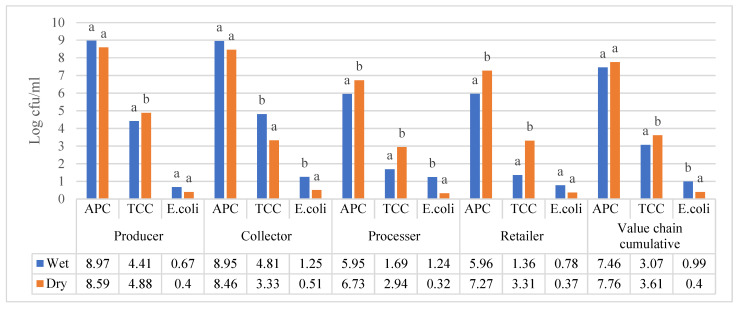
Cumulative APC, TCC, and EC (log cfu/mL) of raw and pasteurized milk value chains located in all study sites. Different superscripts letters above the bar graph are significantly different at (*p* < 0.05).

**Table 1 foods-12-04377-t001:** Percentage of milk and cottage cheese that meet the standard for Total aerobic mesophilic bacteria count (APC) for milk and in the wet and dry seasons.

Region	Quality	Raw Milk		Pasteurized Milk		Cottage Cheese	
		Pass	Fail	Pass	Fail	Pass	Fail
	Log cfu/mL and g	0–6 *n* (%)	>6 *n* (%)	0–4 *n* (%)	>4 *n* (%)	0–4.3 *n* (%)	>4.3 *n* (%)
Oromia	TBC Wet	0 (0%)	48 (100%)	0 (0%)	48 (100%)	0 (0%)	24 (100%)
	TBC Dry	7 (14.58%)	41 (85.42%)	0 (0%)	48 (100%)	0 (0%)	24 (100%)
SNNP	TBC Wet	0 (0%)	24 (100%)	0 (0%)	24 (100%)	0 (0%)	12 (100%)
	TBC Dry	0 (0%)	24 (100%)	0 (0%)	24 (100%)	0 (0%)	12 (100%)
Amhara	TBC Wet	0 (0%)	20 (100%)	0 (0%)	20 (100%)	0 (0%)	8 (100%)
	TBC Dry	0 (0%)	20 (100%)	0 (0%)	20 (100%)	0 (0%)	8 (100%)

**Table 2 foods-12-04377-t002:** Total aerobic mesophilic bacteria count in raw milk, pasteurized milk, and cottage cheese (log cfu/g) in dry and wet seasons.

Study Sites	Season	APC		
		Raw Milk	Pasteurized Milk	Cottage Cheese
Oromia	Wet	8.58 ± 0.37 ^bB^	5.48 ± 0.48 ^aA^	5.17 ± 0.33 ^a^
	Dry	7.41 ± 1.08 ^aB^	6.17 ± 0.63 ^bA^	6.19 ± 0.34 ^b^
SNNP	Wet	9.67 ± 0.52 ^aB^	6.97 ± 0.83 ^aA^	7.34 ± 0.32 ^a^
	Dry	10.80 ± 0.67 ^bB^	9.37 ± 1.65 ^bA^	8.73 ± 1.35 ^b^
Amhara	Wet	8.99 ± 0.19 ^bB^	5.88 ± 0.17 ^aA^	5.69 ± 0.28 ^a^
	Dry	8.46 ± 0.78 ^aB^	6.16 ± 0.73 ^aA^	6.58 ± 0.48 ^b^

Means followed by different capital letter superscripts within the raw values are significantly different from each other between different sample types, and means followed by different small-letter superscripts within the column are significantly different for seasons at (*p* < 0.05).

**Table 3 foods-12-04377-t003:** Percentage of samples that meet the standard for total coliforms count (TCC) established by ESA for milk and cottage cheese during wet and dry seasons.

Regions	Quality	Raw Milk		Pasteurized Milk		Cottage Cheese	
		Pass	Fail	Pass	Fail	Pass	Fail
	Log cfu/mL and g	0–4 *n* (%)	>4 *n* (%)	0–1 *n* (%)	>1 *n* (%)	0–2 *n* (%)	>2 *n* (%)
Oromia	TCC Wet	16 (33.3%)	32 (66.7%)	47 (97.92%)	1 (2.08%)	24 (100%)	0 (0%)
	TCC Dry	11 (22.92%)	37 (77.08%)	4 (8.3%)	44 (91.7%)	10 (41.6%)	14 (58.33%)
SNNP	TCC Wet	13 (54.17%)	11 (45.83%)	5 (20.83%)	19 (79.16%)	11 (91.67%)	1 (8.33%)
	TCC Dry	0 (0%)	24 (100%)	2 (8.3%)	22 (91.7%)	12 (100%)	0 (0%)
Amhara	TCC Wet	1 (5%)	19 (95%)	0 (0%)	20 (100%)	8 (100%)	0 (0%)
	TCC Dry	20 (100%)	0 (0%)	7 (35%)	13 (65%)	8 (100%)	0 (0%)

**Table 4 foods-12-04377-t004:** Comparison of total bacterial count in raw milk, pasteurized milk, and cottage cheese between dry and wet seasons.

Study Sites	Season	TCC		
		Raw Milk	Pasteurized Milk	Cottage Chasse
Oromia	Wet	4.20 ± 2.41 ^aB^	0.04 ± 0.1 ^aA^	3.29 ± 3.18 ^b^
	Dry	5.98 ± 2.04 ^bB^	4.50 ± 2.19 ^bA^	ND ^a^
SNNP	Wet	4.47 ± 1.12 ^aB^	2.67 ± 1.71 ^aA^	0.34 ± 0.81 ^a^
	Dry	3.55 ± 2.43 ^aA^	2.80 ± 2.21 ^aA^	ND ^a^
Amhara	Wet	5.67 ± 0.90 ^bB^	3.71 ± 0.86 ^bA^	ND ^a^
	Dry	2.28 ± 2.48 ^aB^	0.2 ± 0.9 ^aA^	1.59 ± 1.39 ^b^

Means followed by different capital letter superscripts within the raw values are significantly different from each other between different sample types, and means followed by different small letter superscripts within the column are significantly different for seasons at (*p* < 0.05).

**Table 5 foods-12-04377-t005:** Comparison of *Escherichia coli* (log cfu/mL) count in milk and cottage cheese between dry and wet seasons.

Study Sites	Season	*E. coli*		
		Raw Milk	Pasteurized Milk	Cottage Chasse
Oromia	Wet	1.00 ± 1.23 ^bA^	0.9 ± 1.23 ^bA^	NS
	Dry	0.21 ± 0.78 ^aA^	0.12 ± 0.47 ^aA^	NS
SNNP	Wet	1.48 ± 1.48 ^aB^	0.12 ± 0.63 ^bA^	0.34 ± 0.81
	Dry	0.76 ± 1.13 ^aB^	0.13 ± 0.63 ^aA^	NS
Amhara	Wet	0.25 ± 0.65 ^aA^	1.77 ± 1.44 ^aB^	NS
	Dry	0.7 ± 1.28 ^aA^	1.07 ± 1.14 ^aA^	NS

Means followed by different capital letter superscripts for the raw values are significantly different from each other between different sample types, and means followed by different small-letter superscripts within the column are significantly different for seasons at (*p* < 0.05).

**Table 6 foods-12-04377-t006:** Mean comparison of total bacteria, total coliforms, and *E. coli* in cottage cheese samples by regions.

Indicator	Season	Cottage Cheese					
		Oromia		(SNNP)		(Amhara)	
		Producer	Retail	Producer	Retail	Producer	Retail
APC	Wet	5.4 ± 0.26 ^a^	4.9 ± 0.2 ^a^	7.60 ± 0.16 ^a^	7.20 ± 32 ^a^	5.8 ± 28 ^a^	5.59 ± 0.26 ^a^
	Dry	6.05 ± 0.19 ^b^	6.21 ± 0.55 ^b^	8.49 ± 20 ^a^	9.0 ± 51 ^b^	6.28 ± 0.52 ^a^	6.88 ± 0.2 ^b^
TCC	Wet	ND	ND	0.69 ± 1.10 ^a^	ND	ND	ND
	Dry	2.24 ± 2.59 ^b^	4.33 ± 3.48 ^b^	ND	ND	1.08 ± 1.25 ^b^	2.09 ± 151 ^b^
*E. coli*	Wet	ND	ND	ND	ND	ND	ND
	Dry	ND	ND	ND	ND	ND	ND

Means followed by different superscripts within the raw values are significantly different from each other for seasons at (*p* < 0.05).

## Data Availability

All raw data used in the analyses presented here are available in the Appendix A.

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
