# Peer review of "Seasonal Comparison of Microbial Hygiene Indicators in Raw and Pasteurized Milk and Cottage Cheese Collected across Dairy Value Chain in Three Regions of Ethiopia"

_foods, 2023, doi:10.3390/foods12244377_

Round 1

Reviewer 1 Report

Comments and Suggestions for Authors

In the manuscript, the Authors pointed to seasonal changes, but they did not take into account all seasons. This partly misleads the reader. Unfortunately, the samples are random and it is difficult to talk about seasonal differences, when the quality of the raw material and, therefore, the final product depends on very different factors. In the case of raw milk, these are genetic, physiological and environmental factors. In the case of cheese, it is the quality of the raw material and processing, technological issues. There is no standardization of other factors to indicate only seasonal changes. Additionally, 3 regions were also included, which further disturbs the research layout. Moreover, the manuscript is not prepared in accordance with the journal's guidelines.

Comments on the Quality of English Language

The manuscript is written in colloquial language. Vocabulary is not consistent to describe the same issues, e.g. wet season, rainy season, alternately. This misleads the reader. Therefore, an extensive editing of English language is requied.

Reviewer 2 Report

Comments and Suggestions for Authors

The article titled "Seasonal comparison of microbial hygiene indicators in raw and pasteurized milk, and cottage cheese collected across the dairy value chain in three regions of Ethiopia" can be improved in the following sections:

Start with a clear and concise statement about the importance or relevance of the study. Use consistent terminology throughout; e.g., always use "raw milk" instead of sometimes saying "milk". Remove redundancy; you've mentioned the significant variation of hygienic indicator microbial load with season twice. Rather than saying "sustainable awareness creation", it would be clearer to mention what kind of interventions or measures are suggested. Tighten up the background of milk's nutritional value by combining and condensing similar ideas. When discussing the circumstances under which milk is collected, make sure to lead with the impact of these circumstances on milk safety. Clarify the seasonality in Ethiopia before discussing the impact of seasons on microbial load, for logical flow. Try to reduce redundancy in explaining the significance of the study at the end of the introduction.

Clarify the reasons for choosing those specific regions and seasons. Avoid repetition of details about the wet and dry seasons. Be consistent with the number of samples. The abstract mentions 448 samples, but the methodology section lists only 228 samples for the wet season. Differentiate between methods for different bacterial analyses and make sure each step in the method is necessary for the reader to understand. Avoid overly technical explanations and formulae in the main text unless necessary. Consider moving detailed procedures and formulas to an appendix or supplementary material. Clearly state the objective of each test. For instance, before mentioning the t-test, clarify what comparisons are being made. Ensure consistent use of terminology throughout the article. For example, use either "wet season" or "rainy season" consistently, rather than switching between the two. Provide explanations for abbreviations like "APC," "TCC," and "ESA" the first time they are mentioned to make it easier for readers to understand the context. After presenting data, provide a clear interpretation of the results. Explain the significance of the findings and their implications for food safety or dairy production in Ethiopia. Some sentences are quite complex and may benefit from simplification for clarity. Make sure that findings are presented straightforwardly and understandably. Ensure consistency in presenting data throughout the article. For example, make sure that units (e.g., log cfu/ml) are consistently used and formatted. Figures can be used to illustrate trends and comparisons more effectively. Consider adding more figures to visualize the data. The table headings should be clear and concise. Consider revising them to make the information presented in the tables more understandable.

Discuss any limitations of the study, such as sample size or potential sources of bias, to provide a more balanced view of the research. Mention any quality control measures taken during the study to ensure the reliability of the data. The discussion seems to be information-dense with a multitude of references. Breaking down this information into more straightforward points or using tables or charts could help convey this information more efficiently. Ensure that when comparing data from various studies, there is a consistent baseline or parameter for making these comparisons. Every data point or comparison should directly link to the conclusions being made, ensuring that the reader can follow the logical progression. While referencing other studies provides context, ensure that each study cited has direct relevance to the discussion at hand. While the article mentions potential reasons for variations, such as milking and hygienic practices or the presence of a cold chain, delving deeper into why these factors affect the results would provide readers with a better understanding. The discussion identifies various problems, but more emphasis on potential solutions or recommendations for these identified issues might be beneficial. It's crucial to mention if the differences observed are statistically significant, especially when comparing different seasons or methods. While the study focuses on Ethiopia, incorporating broader implications, challenges, or commonalities seen in other regions could enrich the discussion. Considering the length and complexity of the discussion, periodic summaries or recaps could help the reader stay on track and understand the main points being made. The conclusions section should robustly tie together all the major points discussed, emphasizing the importance and implications of the findings, and provide a clear and concise conclusion summarizing the main findings of the study and their implications.

By addressing these points, the article can become more informative, accessible, and impactful for readers.

Comments on the Quality of English Language

Moderate editing of English language required

Reviewer 3 Report

Comments and Suggestions for Authors

Manuscript ID foods-2674746

Seasonal comparison of microbial hygiene indicators in raw and pasteurized milk, and cottage cheese collected across dairy value chain in three regions of Ethiopia

GENERAL REMARKS

Nahusenay et al. analyze seasonal variations in the microbiological hygiene of raw, pasteurized milk and bovine cheese produced in three different areas of Ethiopia. The hygienic-sanitary quality of the dairy products was evaluated through 3 indicators, namely total aerobic mesophilic bacteria, total coliforms, and Escherichia coli counts. In analogy with a large body of literature, the study has considerable value for local agencies to give indications on the level of risk related to the consumption of foods of animal origin that are not correctly managed on a hygienic level. However, unlike other similar investigations, it does not address the causes of contamination at all, limiting itself to describing them exclusively. In my opinion, this could represent a strong limitation of the research. Coming to the manuscript, the introduction is well structured and correctly focuses on the purpose of the research. The same can be said about the discussions and conclusions. On the other hand, I find that the description of the methods and results presentation can be improved.

Below are my recommendations, point by point.

I hope that my suggestions can lead to an improvement in the manuscript.

Regards

SPECIFIC COMMENTS

L 18-19: I'm sorry, but I don't agree with the authors on this statement. As far as I know, the topic of seasonal variation in the microbial load of milk and derived products has already been addressed by numerous studies. I suggest you remove this statement as it stands in its current form, or better rephrase it. Thanks.

L 21: “cottage samples” or cheese cottage samples (as I believe). Please, update. Thanks.

L 23 (and across the manuscript): according to the journal’ template, “p” must be in italics. Thanks.

L 31-32: some keywords are lowercase, and others are uppercase. I ask you to align all keywords to a single style. Thanks.

L 36 (and across the manuscript): according to the journal's template (which I invite you to check), references should be reported in square brackets. Furthermore, I noticed that some references (for example L 38 and 39) have the point (2. and 3.). I ask you to check by eliminating the points in brackets, thanks.

L 44: I find the statement "unclean circumstances" inappropriate in this context. I ask you to paraphrase, thanks.

L 45-47: as far as they are known, what the authors indicate should be referenced in my opinion. Thanks.

L 52: I suggest replacing “mal” with “inappropriate”, thanks.

L 53-55: I find this sentence unclear, in my opinion, due to the omission of verbs. I ask you to paraphrase.

L 61: our survey? Please check, thanks.

L 82 (and across the manuscript): I invite the authors to specify the acronyms at the first mention, thanks.

L 95-100: the study size description, should be improved in my opinion. For any of the investigated areas, authors can detail this part as reported by https://bmcmicrobiol.biomedcentral.com/articles/10.1186/s12866-022-02746-0 at least. Thanks.

L 102-103: what does “purposeful sampling technique” consist of? I ask the authors to clarify this, thanks. If I understand correctly, there is no relationship between the raw milk collected from the producers and the milk sampled later?

L 105: What do the percentages reported for Hawassa (i.e., n = 60%), and Bahir Dar (i.e., n = 48%) refer to? I think there's a mistake. Check, please. Furthermore, I consider it superfluous to report the individual months again, since the survey seasons have already been defined previously.

L 104-112: I think it is obvious that the milk sampled at retailers is pasteurized, but this is not evident at all. Furthermore, since it is pasteurized, it is assumed that there is no microbial load, which however appears in the results. Therefore, I believe appropriate that authors mention the conditions of pasteurization and subsequent storage before sale, which if inadequate may have induced microbial proliferation. Likewise, from the moment milk leaves the dairy farm, the storage and transport conditions should be indicated, in my opinion. For each surveyed area, I believe it appropriate to specify how many of the total seasonal samples are raw milk how many are other types of samples, and whether this attribution is respected between seasons. In addition, there is no mention of the type of cheese analyzed, the production process, and the stage of the supply chain at which it was sampled (farm, retailer, etc.). I ask you to update these points.

L 116: the cited reference doesn’t fit either the suggestion reported for L 82 or for L 36. Authors are requested to update, thanks.

L 141: the previous comment also applied to “3M Food Safety, 2017” method. Thanks.

L 173 (and others): in my opinion, it would be appropriate not to start any sentence with an acronym. Thanks.

L 175: the use of references 25 and 26 with respect to table 1 is not clear. Do the values in the table derive from the standards indicated by the aforementioned references? I ask you to check, thanks.

L 178: check the reference style for “Mengistu et al., 2023”, thanks.

L 184-188 (as well as for other result sections): I find it very redundant to report in the text the same data clearly listed in the table. I advise you to consider the possibility of re-modulating this part, thanks.

L 185: the way of indicating table 2 (i.e., table (2)) is unusual. Is there a particular reason? Explain it, please.

L 394: I don't notice any table 9. Please check it out

Round 2

Reviewer 2 Report

Comments and Suggestions for Authors

The discussion section is still information-dense with a multitude of references, which could be improved by breaking down the information into more straightforward points or using tables or charts for more efficient conveyance of information. Additionally, there is a suggestion to provide more emphasis on potential solutions or recommendations for the identified issues discussed in the article and to mention if the observed differences are statistically significant, particularly when comparing different seasons or methods. Lastly, it is recommended to incorporate broader implications, challenges, or commonalities seen in other regions to enrich the discussion.

The response indicates that these suggestions must be addressed before further proceedings.

Comments on the Quality of English Language

Minor editing of English language required

Author Response

Reviewer Comment: The discussion section is still information-dense with a multitude of references, which could be improved by breaking down the information into more straightforward points or using tables or charts for more efficient conveyance of information.

Response: Thanks for the constructive comment. We have included an additional table (i.e. Table 7) that summarized a comparison between the present study results some of the data literatures available in Ethiopia for more efficient conveyance of information.. We also tried to break down the discussion dividing the paragraphs based on the chronological order of the value chains (raw milk, pasteurized milk and cottage cheese) that we believe ease understanding the ideas. 

Reviewer Comment: Additionally, there is a suggestion to provide more emphasis on potential solutions or recommendations for the identified issues discussed in the article.

Response: We are currently preparing another survey research manuscript that was parallelly collected with milk samples that will discuss in more detail the reasons for such milk contamination across the milk value chain with several specific recommendations and the volume of these paper didn’t allow us to mix the two information together here. But we believe this paper has several recommendations in the conclusion section. Moreover, our research team have other published papers that discussed the key risk factor and potential solutions. Here we outlined for you the other papers for your reference.

1)            Abdi Bedassa, Henok Nahusenay, Zerihun Asefa, Tesfaye Sisay, Gebrerufael Girmay, Jasna Kovac, Jessie L. Vipham and Ashagrie Zewdu (2023). Prevalence and associated risk factors for Salmonella enterica contamination of cow milk and cottage cheese in Ethiopia. Food Safety and Risk. fe7c7527-efd2-4e93-97b5-1eb9dc8804d9   |  v.2.1

2)            Abera Admasie, Adane Eshetu, Tesfaye Sisay Tessema, Jessie Vipham, Jasna Kovac and Ashagrie Zewdu (2023), Prevalence of Campylobacter species and associated risk factors for contamination of dairy products collected in a dry season from major milk sheds in Ethiopia (2023). Food Microbiology. 109 (2023) 104145.

3)            Betelhem Mengstu, Alganesh Tola, Henok Nahusenay, Tesfaye Sisay, Jasna Kovac, Jessie Vipham, Ashagrie Zewdu (2023).  Evaluation of microbial hygiene indicators in raw milk, pasteurized milk and cottage cheese collected across the dairy value chain in Ethiopia. International Dairy Journal 136 (2023) 105487.

Reviewer 3 Report

Comments and Suggestions for Authors

Dear authors,

I have evaluated the revised version of manuscript foods-2674746. I have noted your extensive revision, which meets my requirements. As a final suggestion, I ask you to update the text to line 464. In fact, the authors use the statement 'according to our survey', claiming that the results of this survey have already been published or will be published in the future. Well, if already published, I would ask you to add the reference; conversely, if unpublished, it would be appropriate to add the statement "unpublished data". Thanks

Author Response

Reviewer 3 Comment: As a final suggestion, I ask you to update the text to line 464. In fact, the authors use the statement 'according to our survey', claiming that the results of this survey have already been published or will be published in the future. Well, if already published, I would ask you to add the reference; conversely, if unpublished, it would be appropriate to add the statement "unpublished data". Thanks

Response: Thank you very much for the comment and we have now added the statement "unpublished data" on line 464 and all the others places that we mentioned about our survey result. This comment has reminded us that we should work on our survey results after this as it answers the "why" and we will do our best to get the survey results get published as well in the near future. Thanks